systems biology/computational biology

immuno-oncology, immune checkpoint inhibitor, computational model, systems biology, computational biology

**Author for correspondence:**
Hanwen Wang
e-mail: hwang163@jhu.edu

# *In silico* simulation of a clinical trial with anti-CTLA-4 and anti-PD-L1 immunotherapies in metastatic breast cancer using a systems pharmacology model

Hanwen Wang[1], Oleg Milberg[1], Imke H. Bartelink[2,3,4], Paolo Vicini[5], Bing Wang[6], Rajesh Narwal[7], Lorin Roskos[7], Cesar A. Santa-Maria[8] and Aleksander S. Popel[1,8]

[1]Department of Biomedical Engineering, School of Medicine, Johns Hopkins University, Baltimore, MD 21205, USA
[2]Department of Medicine, University of California, San Francisco, CA, USA
[3]Clinical Pharmacology, Pharmacometrics and DMPK (CPD), MedImmune, South San Francisco, CA, USA
[4]Department of Clinical Pharmacology and Pharmacy, Amsterdam UMC, Vrije Universiteit Amsterdam, The Netherlands
[5]Clinical Pharmacology, Pharmacometrics and DMPK, MedImmune, Cambridge, UK
[6]Amador Bioscience Inc, Pleasanton, CA 94588, USA
[7]Clinical Pharmacology and DMPK (CPD), MedImmune, Gaithersburg, MD, USA
[8]Department of Oncology and Sidney Kimmel Comprehensive Cancer Center, Johns Hopkins University, Baltimore, MD, USA

HW, 0000-0001-5480-431X

The low response rate of immune checkpoint blockade in breast cancer has highlighted the need for predictive biomarkers to identify responders. While a number of clinical trials are ongoing, testing all possible combinations is not feasible. In this study, a quantitative systems pharmacology model is built to integrate immune–cancer cell interactions in patients with breast cancer, including central, peripheral, tumour-draining lymph node (TDLN) and tumour compartments. The model can describe the immune suppression and evasion in both TDLN and the tumour microenvironment due to checkpoint expression, and mimic

the tumour response to checkpoint blockade therapy. We investigate the relationship between the tumour response to checkpoint blockade therapy and composite tumour burden, PD-L1 expression and antigen intensity, including their individual and combined effects on the immune system, using model-based simulations. The proposed model demonstrates the potential to make predictions of tumour response of individual patients given sufficient clinical measurements, and provides a platform that can be further adapted to other types of immunotherapy and their combination with molecular-targeted therapies. The patient predictions demonstrate how this systems pharmacology model can be used to individualize immunotherapy treatments. When appropriately validated, these approaches may contribute to optimization of breast cancer treatment.

# 1. Introduction

Immune checkpoint blockade including antibodies targeting cytotoxic T lymphocyte antigen-4 (CTLA-4) and programmed death-1 (PD-1) proteins, have demonstrated anti-cancer activity in multiple cancer types [1]. Breast cancer is the second leading cause of cancer-related death in women, and despite the need for better therapies, has shown limited response to immune checkpoint therapy [2,3]. Numerous clinical trials of checkpoint blockade combined with novel agents have consistently demonstrated response rates in metastatic breast cancer ranging from 4 to 59% in triple-negative breast cancer (TNBC) [4–7]. The wide range of efficacy of checkpoint blockade therapy in breast cancer reflects an unmet need for predictive biomarkers that can help identify potential responders to the therapy. Studies of breast tumour microenvironment (TME), and specifically tumour immune microenvironment (TIME) provide a better understanding of the immune evasion mechanisms in breast tumours and help identify potential biomarkers and targets for future therapies [8].

Breast cancer cells have several mechanisms of immune evasion, which help them survive the attack by cytotoxic T lymphocytes (CTLs). Among these mechanisms, the checkpoint interactions are identified to be one of the key mechanisms, and thus the expression of PD-L1 on tumour cells is considered to be a predictive biomarker [9]. Although the interaction between PD-1 and its ligand PD-L1 inhibits CTLs proliferation and effector function, PD-L1 upregulation correlates to higher metastasis-free and overall survival rate and higher response rate to chemotherapy in TNBC [10]. In recent clinical studies, high PD-L1 expression on tumour-infiltrating immune cells also correlates to higher survival rates to both anti-PD-L1 monotherapy and combination therapy with nanoparticle albumin-bound (nab)-paclitaxel in metastatic TNBC [11,12]. The better survival and response rate are possibly due to the fact that high PD-L1 expression on tumour cells reflects a high level of tumour-infiltrating lymphocytes (TILs), which serves as a negative feedback mechanism. Thus, in concurrent treatment with chemotherapy, PD-L1 blockade is expected to restore effector function of CD8+ CTLs in TME, resulting in an even higher response rate in PD-L1 upregulated subtypes [13].

Another possible factor that leads to the limited efficacy of immune checkpoint blockade therapy is the low mutational load of breast cancer [14,15]. To eradicate a tumour, highly immunogenic antigens expressed by tumour cells are required to trigger T-cell activation and recognition. Multiple studies have indicated that high non-synonymous mutation burden in non-small cell lung cancer (NSCLC) correlates to a better objective response and progression-free survival rate in anti-PD-1/PD-L1 therapy [16,17]. In a PD-1 blockade therapy of NSCLC using pembrolizumab, the cohort with low mutational burden had an overall response rate (ORR) of 0%. Although the correlation between mutational load in breast cancer and its response to immunotherapy is still under investigation, the low mutation frequency of breast cancer may be responsible for the limited efficacy of checkpoint blockade therapy [14]. In fact, TNBC, which has significantly higher mutational load than non-TNBC, has consistently higher complete response rates to neoadjuvant chemotherapy than hormone receptor-positive breast therapy [4,5]. This correlation between mutational load and response rate makes it a possible biomarker of interest in checkpoint blockade therapy.

Although the correlation between individual factors above and tumour response to immunotherapy is well characterized in some cancer types, their combined effects remain unclear. PD-L1 expression has not consistently correlated with response to immunotherapy; likely because immune–cancer cell interactions are exceedingly complex and not captured by a single biomarker [1,18]. Computational systems pharmacology models may be able to integrate the numerous interacting parts and processes and help unravel the complexity of cancer and the immune system [19]. This study proposes a systems pharmacology model including compartments that together represent comprehensive cancer–immune

cell interactions in patients with breast cancer. This model is our first attempt to integrate various compartments involved in cancer–immune cell interactions, including tumour-draining lymph node and tumour microenvironments, to study the effects of checkpoint blockade therapy, which can help improve the screening and immunotherapeutic strategies in clinical trials.

The proposed model aims to identify potential predictive biomarkers, hypothesize the possible immune evasion mechanism, and eventually predict the response of patient cohorts or even individual patients to immunotherapy. In this study, we will focus on the combination therapy using durvalumab and tremelimumab using the dosing regimen in a clinical trial of metastatic breast cancer by Santa-Maria *et al.* [20]. Tremelimumab is an anti-CTLA-4 monoclonal antibody (mAb) that blocks CTLA-4 interaction with CD80/86, and durvalumab is an anti-PD-L1 mAb that blocks PD-L1 interaction with PD-1 and CD80 [21,22]. In this pilot study of durvalumab and tremelimumab in metastatic breast cancer, 18 evaluable patients were enrolled, and 75 mg tremelimumab was administered with 1500 mg durvalumab monthly for four cycles followed by 750 mg durvalumab monotherapy every two weeks for up to 2 years. Among the 18 patients who were eligible for primary analysis, 11 TNBC and 7 estrogen receptor-positive (ER+) patients exhibited overall response rates of 43% and 0%, respectively.

# 2. Methods

## 2.1. Model overview and cell dynamics

The quantitative systems pharmacology (QSP) model consists of four compartments: central, peripheral, tumour and tumour-draining lymph node (TDLN). Central and peripheral compartments represent the total volume of blood and peripheral tissues, respectively. Tumour compartment represents the total tumour volume, which is assumed to be constant for the purpose of antibody pharmacokinetics and effector T-cell transport. TDLN compartment represents a lumped lymph node assuming that the antibody will be evenly distributed among multiple TDLNs that have the same antibody and T-cell dynamics. The model comprises 275 ordinary differential equations (ODEs) and 206 algebraic equations and is implemented using SimBiology toolbox in MATLAB (MathWorks, Nathick, MA). Figure 1*a* illustrates the dynamics of major species in the model. To ensure reproducibility of the model, the complete set of governing ODEs, model parameters and SBML code are presented in the electronic supplementary material.

## 2.2. Immune activation

The immune response is initiated from the tumour where neoantigens are produced. Neoantigens can be either engulfed by antigen-presenting cells (APCs) via phagocytosis in the tumour or transported into the lymph node via lymphatic vessels, where they can be taken up by resident APCs in TDLNs [18]. Mature APCs (mAPCs) in the tumour can migrate back to the lymph node via lymphatic vessels and there initiate T-cell priming and activation to produce effector T cells [23]. The effector T cells then intravasate into the circulatory system and are transported to the tumour where they extravasate into the TME and where they kill tumour cells via cytotoxic activity [24]. Killed tumour cells can release more neoantigens, which will further promote maturation of APCs and activation of effector T cells, forming a feedforward loop until all tumour cells are eradicated [25]; alternatively, the process may stabilize or even reverse depending on the systems parameters and dynamics (e.g. a rapidly growing tumour may overcome the immune stress and continue to grow albeit at lower rate than in the absence of the feedforward loop).

## 2.3. Immune suppression

Regulatory T cells (Tregs) and myeloid-derived suppressor cells (MDSCs), which contribute to the immunosuppressive tumour microenvironment, are included in the tumour compartment [26]. The number of MDSC is estimated by the number of cancer cells in the tumour, and the number of Tregs in TDLN and tumour compartment are estimated by the total number of T lymphocytes in lymph node and as a function of MDSC level in the tumour, all based on the literature evidence [27]. Their inhibitory functions are implemented through checkpoint expression including PD-1, PD-L1 and CTLA-4. The inhibitory effects of Tregs and MDSCs on effector T-cell activity and mAPC maturation

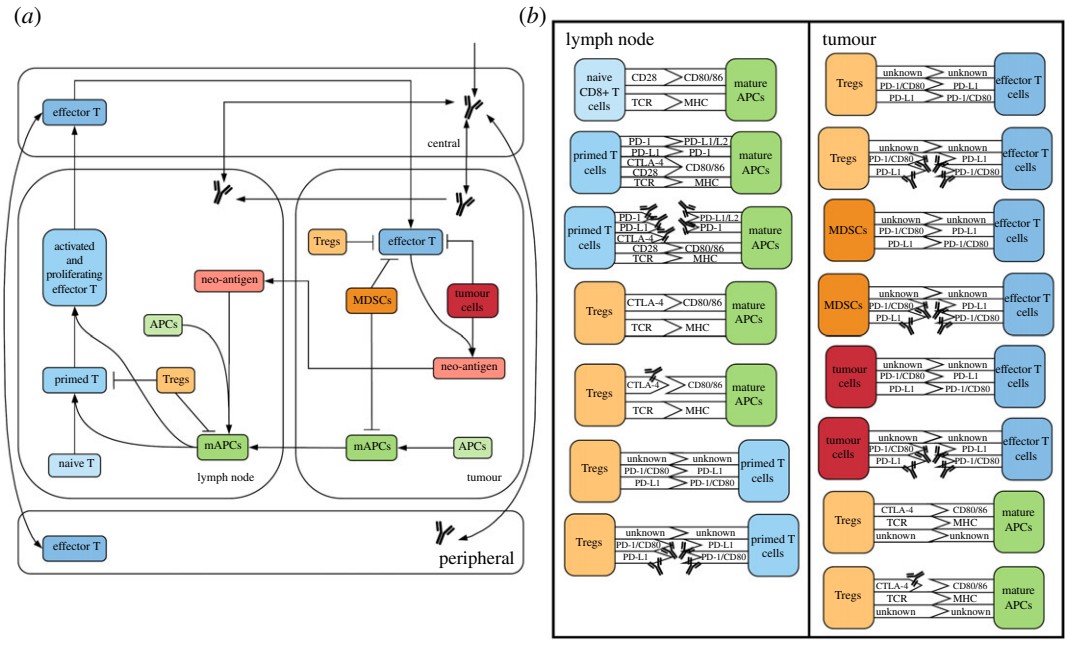

**Figure 1.** Diagram of model. (*a*) Diagram of all the molecular and cellular dynamics in each compartment. (*b*) Diagram of all ligand–receptor interactions in the model, focusing on immune checkpoints and their blockade by antibody treatment using anti-PD-1/PD-L1/CTLA-4 antibodies.

are partially due to the checkpoint expression, calculated by receptor occupancies and a limiting factor describing the maximum inhibitory effect. In addition, tumour cells are able to partially inhibit CTL proliferation and cytotoxic activity, either transiently or permanently, via checkpoint expression of PD-L1 and PD-L2 [28]. All the ligand–receptor interactions are illustrated in figure 1*b*. While all ligands and receptors are expressed on cell surfaces, whose interactions occur within the immunological synapses, CTLA-4 can also be secreted and released from Treg cell surfaces into the TME, where it can bind CD80/86 receptors on mAPCs [29–31]. These inhibitory signals suppress the feedforward loop of cancer-killing and T-cell activation in the host immune system.

## 2.4. T-cell priming

All the ligand–receptor interactions are implemented using two steps. First, the engagement between two cell types forms a cell–cell complex. Then, the cell–cell complex can either dissociate without any effect or results in the activation/inhibition of either cell. This decision is dependent on the checkpoint signals, calculated by receptor occupancies or antigen intensity, which refers to the binding affinity between the T cell receptor (TCR) and neoantigen-specific MHC. For example, the T-cell priming occurs in three distinct stages in TDLNs [32]. Naive T cells first engage in transient interactions with mAPCs to become primed naive T cells, followed by stable contacts inducing constant chemokine production to become proliferating T cells, and finally become effector T cells with high motility and rapid proliferation [32]. Each time the interaction between mAPCs and T cells occurs, the two cells can either disengage without activation of naive T cell or disengage with naive T cells moving on to the next stage, which depends on both antigen intensity and PD-L1 receptor occupancy.

## 2.5. T-cell transport

Once T cells are produced from TDLNs, they are transported into the tumour compartment through the circulatory system. The T-cell transport is implemented as a three-step mechanism: reversible attachment of free CTLs onto the vascular wall, irreversible adhesion of attached CTLs and transmigration into the extracellular space, adopted from the physiologically based kinetic model by Zhu *et al.* [24]. The T-cell migration from central to both peripheral and tumour compartments is

described by the following ODEs:

$$\frac{\mathrm{d}(\mathrm{Effector\,Tf})}{\mathrm{d}t} = -k_f \cdot \left(B_T - \frac{\mathrm{EffectorTB}}{V_v} - \frac{\mathrm{EffectorTa}}{V_v}\right) \cdot \mathrm{EffectorTf} + k_r \cdot \mathrm{EffectorTb} + \frac{Q}{V_v}$$
$$\cdot \mathrm{EffectorTB} - \frac{Q - k_L}{V_v} \cdot \mathrm{EffectorTf},$$

$$\frac{\mathrm{d}(\mathrm{EffectorTb})}{\mathrm{d}t} = -Eff\,T_{\mathrm{Turnover}} \cdot \mathrm{EffectorTb} + k_f \cdot \left(B - \frac{\mathrm{EffectorTB}}{V_v} - \frac{\mathrm{EffectorTa}}{V_v}\right) \cdot \mathrm{EffectorTf}$$
$$- k_r \cdot \mathrm{EffectorTb} - AR \cdot \mathrm{EffectorTb}$$

and $$\frac{\mathrm{d}(\mathrm{EffectorTa})}{\mathrm{d}t} = -J \cdot \mathrm{EffectorTa} - Eff\,T_{\mathrm{Turnover}} \cdot \mathrm{EffectorTa} + AR \cdot \mathrm{EffectorTb}.$$

Here EffectorTB is the number of T cells in the central (blood) compartment, EffectorTf is the number of free T cells in the vascular space, EffectorTb is the number of captured T cells in the vascular space, and EffectorTa is the number of arrested T cells in the vascular space. Transport parameters, $k_f$ and $k_r$, attachment and detachment rate, $k_L$, lymphatic transport rate, AR, adhesion rate, $J$, transmigration rate, $B$, *ad hoc* adhesion site density, $Vv$, vascular space volume, $Q$, blood flow rate, and $EffT_{\mathrm{Turnover}}$, cell death rate, are estimated based on the literature data [24,33].

## 2.6. Pharmacokinetic/pharmacodynamic model of antibody

There are published population pharmacokinetic (PK) models of tremelimumab and durvalumab built using pharmacokinetic data from clinical trials [34,35]. The PK models were rebuilt with the same parameters using SimBiology toolbox in MATLAB, and the model predictions of antibody plasma concentration with different doses were recorded to optimize the kinetic parameters in our model using pattern search in the Global Optimization Toolbox (electronic supplementary material, figure S5). For example, the ODEs governing the PK/pharmacodynamic (PD) of tremelimumab between compartments are described as follows:

$$\frac{\mathrm{d}(CTLA4_{mabB})}{\mathrm{d}t} = \frac{1}{V_{\mathrm{Blood}}} \left(\frac{k_{DoseAd\,\min,AntiCTLA4} \cdot \mathrm{BodyWeight} \cdot CTLA4mAb}{\mathrm{Tremelimumab}_{MW} \cdot V_{\mathrm{Blood}}} \cdot V_{\mathrm{Blood}} - Cl_{CTLA4}\right.$$
$$\cdot CTLA4_{mabB} - \left(\frac{k_{CTLA4,BP} \cdot SA_{BP}}{K_{B,CTLA4}}\right) \cdot CTLA4_{mabB} \cdot V_{\mathrm{Peripheral}}$$
$$+ \left(\frac{k_{CTLA4,BP} \cdot SA_{BP}}{K_{P,CTLA4}}\right) \cdot CTLA4_{mabP} \cdot V_{\mathrm{Peripheral}} - \left(\frac{k_{Ab,BT} \cdot SA_{BT}}{K_{B,CTLA4}}\right)$$
$$\cdot CTLA4_{mabB} \cdot V_{\mathrm{Tumour}} + \left(\frac{k_{Ab,BT} \cdot SA_{BT}}{K_{T,CTLA4}}\right) \cdot CTLA4_{mabT} \cdot V_{\mathrm{Tumor}}$$
$$+ \left(k_{Lt} \cdot \frac{V_{\mathrm{Tumour}}}{V_{td\ln} \cdot K_{LN}} + \frac{k_{Ab,BLN} \cdot SA_{BP}}{K_{LN}} \cdot \mathrm{Num}_{TDLN}\right) \cdot V_{td\ln} \cdot CTLA4_{mab}$$
$$\left. - \frac{k_{Ab,BLN} \cdot SA_{BP}}{K_{B,CTLA4}} CTLA4_{mabB} \cdot V_{td\ln} \cdot \mathrm{Num}_{TDLN}\right.$$

$$\frac{\mathrm{d}(CTLA4_{mabP})}{\mathrm{d}t} = \frac{1}{V_{\mathrm{Peripheral}}} \left(\left(\frac{k_{CTLA4,BP} \cdot SA_{BP}}{K_{B,CTLA4}}\right) \cdot CTLA4_{mabB} \cdot V_{\mathrm{Peripheral}}\right.$$
$$\left. - \left(\frac{k_{CTLA4,BP} \cdot SA_{BP}}{K_{B,CTLA4}}\right) \cdot CTLA4_{mabP} \cdot V_{\mathrm{Peripheral}}\right)$$

$$\frac{\mathrm{d}(CTLA4_{mabT})}{\mathrm{d}t} = \frac{1}{V_{\mathrm{Tumour}}} \left(k_{Ab,BT} \cdot \frac{SA_{BT}}{K_{B,CTLA4}} \cdot CTLA4_{mabB} \cdot V_{\mathrm{Tumour}} - k_{Ab,BT} \cdot \frac{SA_{BT}}{K_T} \cdot CTLA4_{mabT}\right.$$
$$\cdot V_{\mathrm{Tumour}} - k_{Lt} \cdot \frac{V_{\mathrm{Tumour}}}{K_T} \cdot CTLA4_{mabT} - k_{\mathrm{on},CTLA4mAb-CTLA4} \cdot CTLA4_{TregT}$$
$$\cdot CTLA4_{mabT} + k_{\mathrm{off},CTLA4mAb-CTLA4} \cdot [CTLA4{:}CTLA4_{TrT}] - k_{\mathrm{on},CTLA4mAb-CTLA4}$$
$$\left. \cdot CTLA4_{mabT} \cdot CTLA4_{TregTS} + k_{\mathrm{off},CTLA4mAb-CTLA4} \cdot [CTLA4{:}aCTLA4_{TrTS}]\right)$$

$$\frac{\mathrm{d}(CTLA4_{mab})}{\mathrm{d}t} = \frac{1}{V_{td\ln}}\left(k_{Ab,BLN} \cdot \frac{SA_{BP}}{K_{B,CTLA4}} \cdot CTLA4_{mabB} \cdot V_{td\ln} + k_{Lt} \cdot \frac{V_{\mathrm{Tumour}}}{K_T}\right.$$

$$\cdot \frac{CTLA4_{mabT}}{\mathrm{Num}_{TDLN}}$$

$$- \left(k_{Lt} \cdot \frac{V_{\mathrm{Tumour}}}{V_{td\ln} \cdot K_{LN} \cdot \mathrm{Num}_{TDLN}} + k_{Ab,BLN} \cdot \frac{SA_{BP}}{K_{LN}}\right) \cdot V_{td\ln} \cdot CTLA4_{mab}$$

$$- k_{\mathrm{on},CTLA4mAb-CTLA4} \cdot CTLA4_{mab} \cdot PNT_{CTLA4} + k_{\mathrm{off},CTLA4mAb-CTLA4}$$

$$\cdot [CTLA4_{mAb}{:}CTLA4] - k_{\mathrm{on},CTLA4mAb-CTLA4} \cdot [Tr{:}mAPC_{CTLA4}] \cdot CTLA4_{mab}$$

$$+ k_{\mathrm{off},CTLA4mAb-CTLA4} \cdot [TrALN{:}CT{:}aCT] - k_{\mathrm{on},CTLA4mAb-CTLA4} \cdot [TrLN_{CTLA4}]$$

$$\cdot CTLA4_{mab} + k_{\mathrm{off},CTLA4mAb-CTLA4} \cdot [TrLN{:}CT{:}aCT] - k_{\mathrm{on},CTLA4mAb-CTLA4}$$

$$\left. \cdot [TrLN_{CTLA4S}] \cdot CTLA4_{mab} + k_{\mathrm{off},CTLA4mAb-CTLA4} \cdot [TrLN{:}CTLA4S{:}aCTLA4S]\right)$$

Here $CTLA4_{mabB}$, $CTLA4_{mabP}$, $CTLA4_{mabT}$ and $CTLA4_{mab}$ refer to the concentration of tremelimumab in central, peripheral, tumour and TDLN compartment, respectively. $V_{\mathrm{Blood}}$, total blood volume, $V_{\mathrm{Peripheral}}$, total peripheral volume, $V_{\mathrm{Tumour}}$, constant tumour volume for antibody PK only, $V_{tdln}$, volume of each TDLN, $SA_{BP}$, surface area to volume ratio between the central and the peripheral and TDLN compartments, $SA_{BT}$, surface area to volume ratio between blood and tumour compartments, $K_T$, available volume fraction in the tumour, $K_{LN}$, available volume fraction in each TDLN, $k_{CTLA4,BP}$, permeability to antibody between blood and peripheral compartments, $k_{Ab,BT}$, permeability to antibody between blood and tumour compartments, $k_{Ab,BLN}$, permeability to antibody between blood and TDLN compartments, and $k_{Lt}$, lymph flow rate, are estimated based on literature data assuming that large antibody proteins of similar sizes have the same permeability between compartments and that surface area between TDLN and other compartments equals to $SA_{BP}$ [36,37]. $Cl$, clearance rate, $K_B$, available volume fraction in the blood, and $K_P$, available volume fraction in the peripheral compartment, are optimized based on the published population PK models [34,35]. The estimation of permeability and surface area to volume ratio is explained in detail in the electronic supplementary material. As for the PD of tremelimumab, $k_{\mathrm{on},CTLA4mAb-CTLA4}$ and $k_{\mathrm{off},CTLA4mAb-CTLA4}$, association and dissociation rate of tremelimumab with CTLA-4 are estimated based on experimental data [38]. $CTLA4_{\mathrm{Treg}T}$, CTLA-4 expressed on Treg surface in the tumour, $CTLA4_{\mathrm{Treg}TS}$, soluble CTLA-4 secreted by Treg in the tumour, $PNT_{CTLA4}$, CTLA-4 expressed on primed T-cell surface in TDLNs, $Tr{:}mAPC_{CTLA4}$, CTLA-4 expressed on Treg interacting with mAPCs in TDLNs, $TrLN_{CTLA4}$, CTLA-4 expressed on non-interacting Treg surface in TDLNs, $TrLN_{CTLA4S}$, soluble CTLA-4 secreted by Treg in TDLNs, and the corresponded complexes when bound with tremelimumab, $CTLA4{:}CTLA4_{TrT}$, $CTLA4{:}aCTLA4_{TrTS}$, $CTLA4_{mAb}{:}CTLA4$, $TrALN{:}CT{:}aCT$, $TrLN{:}CT{:}aCT$ and $TrLN{:}CTLA4S{:}aCTLA4S$ are further specified, along with the values of other parameters for both antibodies, in electronic supplementary material, table S2.

## 2.7. Simulation settings

The model is used to simulate PK/PD for anti-CTLA-4, anti-PD-1 and anti-PD-L1 antibodies in monotherapy and combination therapy. Since this study focuses on a specific clinical trial, namely on combination therapy for 18 breast cancer patients using tremelimumab and durvalumab, the parameters, including the number of tumour-draining lymph nodes, tumour growth rate, checkpoint expression and cancer cell diameter, are estimated to be metastatic breast cancer-specific [20]. The values and ranges of parameters with the references are presented in the electronic supplementary material, table S6, together with the complete governing equations, model parameters, as well as SBML code. Figure 2 demonstrates the main outputs of the model: time-dependent tumour size change from the start of therapy, total number of effector T cell originating from the lymph node and the number of mature APCs in the lymph node. Simulations are performed by setting (a) a tumour diameter at the beginning of the therapy, which is used to calculate the initial tumour volume, (b) antigen intensity, and (c) PD-L1 expression, which refers to the percentage of tumour cells expressing PD-L1. Although the heterogeneity of spatial distribution in each compartment is not considered in this study, the checkpoint expression on cancer cells can be heterogeneously distributed. Cancer cells are divided into four 'subtypes' in the tumour compartment: cells that do not express any checkpoint; and cells that express PD-L1 only, or PD-L2 only, or express both. The PD-L1 expression and a constant PD-L2 expression are used as the probability of a cancer cell expressing each checkpoint to calculate the number of cancer cells in each 'subtype'. Expression of CD80 and PD-1 on cancer cells is also implemented for use on other cancer types but is set to be zero for breast cancer.

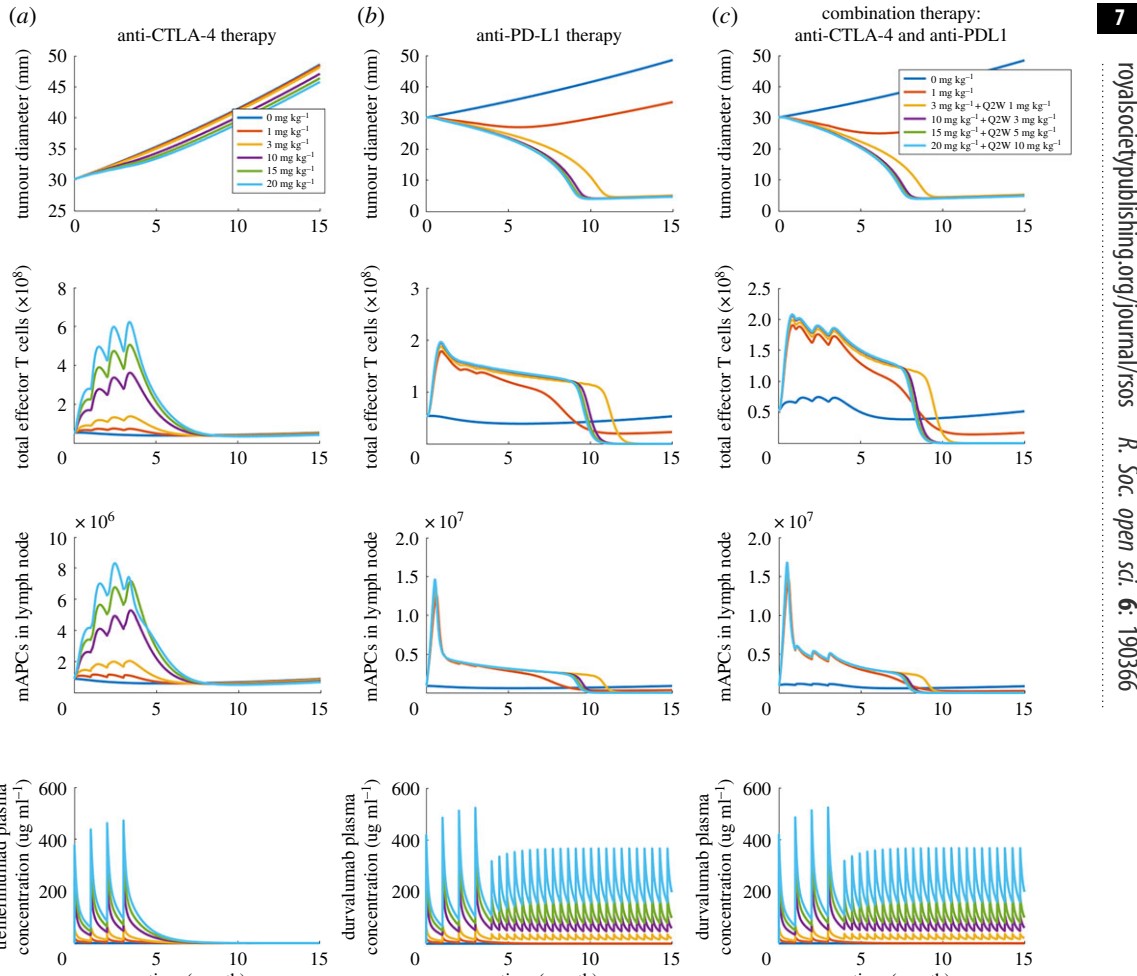

**Figure 2.** Outputs of the model with different doses and regimen. Time-dependent variables including tumour diameter, total effector T cells in the system, mAPCs in lymph node, and corresponding antibody serum concentration in (*a*) anti-CTLA-4 therapy, (*b*) anti-PD-L1 therapy and (*c*) combination therapy.

Baseline starting tumour size, antigen intensity and PD-L1 expression are assumed to be 30 mm, 0.7 and 25%, respectively, which are selected to best illustrate the heterogeneity of tumour response and to investigate the sensitivity of each parameter of interest. The steady-state and dynamic solutions are calculated using the Sundials solver. The absolute tolerance and relative tolerance are set to be $10^{-14}$ (day) and $10^{-13}$, respectively. Within a time period typically on the order of two months, the number of effector T cells reaches a quasi-steady state, because the T-cell activation is balanced by the inhibition of effector T cells in the TME. Once the steady state is reached, the antibody is administered into the central compartment through intravenous infusion, and the simulation is continued. Tumour growth is simulated for 15 months after therapy begins.

## 3. Results

### 3.1. Effect of combination of CTLA-4 and PD-L1 blockade

Since the model aims at predicting patients' responses to immunotherapies, we define the main outputs to be: analysis of time-dependent tumour size change, effector T-cell production and antigen-presenting cell maturation in TDLNs. To illustrate the predictions, we first consider a baseline case and show the dose response to each monotherapy, anti-CTLA-4 and anti-PD-L1, and to their combination, following the regimen of the clinical trial [20]. In characterizing tumour growth, we use RECIST criteria [39], as was done in the clinical trial. The main outputs of 'virtual' patients (using baseline parameters) with a starting tumour size of 30 mm at the beginning of therapies are shown in figure 2 for various doses of checkpoint blockade antibodies. Assuming an average body weight of 75 kg in the model,



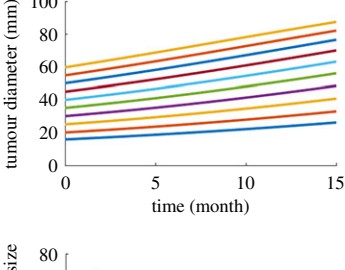

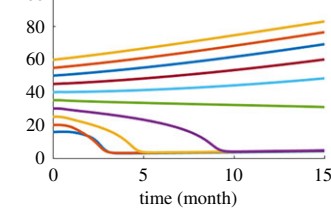

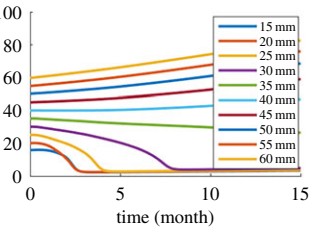

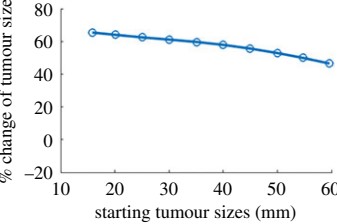

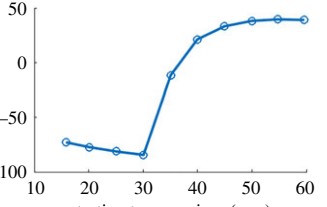

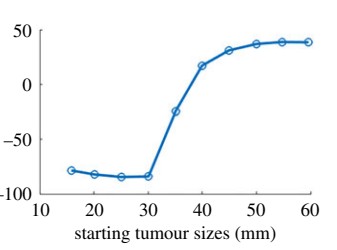

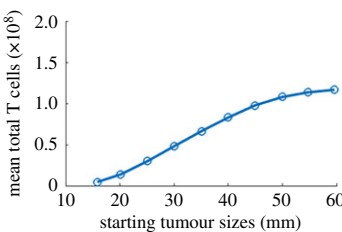

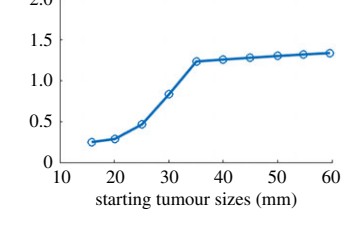

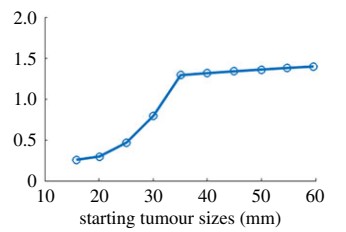

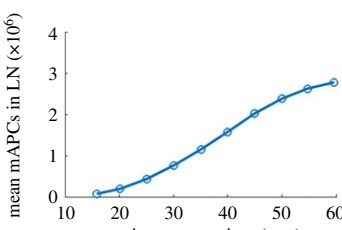

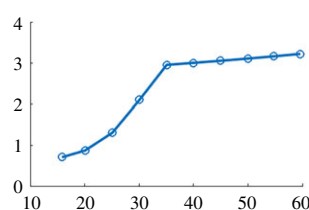

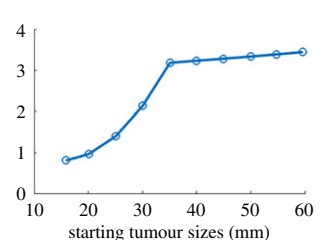

**Figure 3.** ($a$–$c$) Time-dependent tumour diameter, the percentage change of end tumour size, mean total T cells and mean mAPC in the lymph node with different starting tumour sizes from the time of therapy to the 15th month after therapy begins.

tremelimumab is administrated monthly four times at doses of 0, 1, 3, 10, 15, 20 mg kg$^{-1}$ in anti-CTLA-4 monotherapy (figure 2$a$). While the monotherapy causes a significant increase of effector T cells and mature antigen-presenting cells (mAPCs), based on the tumour size, patients show progressive disease, per RECIST criteria, for doses of 0, 1, 3 mg kg$^{-1}$, and for higher doses of 10, 15, 20 mg kg$^{-1}$ stable disease for about five months followed by progressive disease. For anti-PD-L1 monotherapy, durvalumab is administrated monthly four times at doses of 0, 1, 3, 10, 15, 20 mg kg$^{-1}$, where 3, 10, 15, 20 mg kg$^{-1}$ doses are followed by 1, 3, 5, 10 mg kg$^{-1}$ every two weeks up to 15 months, respectively. Although anti-PD-L1 monotherapy did not increase T-cell production and APC maturation as much as anti-CTLA-4 monotherapy, patients show responses with four doses of 1 mg kg$^{-1}$ and higher (figure 2$b$). When durvalumab is combined with a fixed dose of 1 mg kg$^{-1}$ tremelimumab, T-cell production is increased by three- to fourfold compared to the beginning of the therapy, which is higher than that in anti-PD-L1 monotherapy (figure 2$c$). The results suggest that the PD-1/PD-L1 pathway plays a vital role in the TME as a resistance mechanism to protect tumour cells from tumour-infiltrating lymphocytes. In combination therapy, the upregulation of T-cell production by CTLA-4 blockade in TDLNs is observed, and the effect on tumour size is moderately enhanced, compared to the anti-PD-L1 monotherapy.

## 3.2. Factors influencing tumour response to immunotherapy

To investigate the effects of starting tumour size, PD-L1 expression and antigen intensity on tumour response, we plot the time-dependent tumour size, percentage change of tumour size with respect to the starting tumour size, average number of effector T cells and mAPCs in TDLN over the 15 months by changing one parameter at a time in figures 3–5. Four doses of 1 mg kg$^{-1}$ tremelimumab are

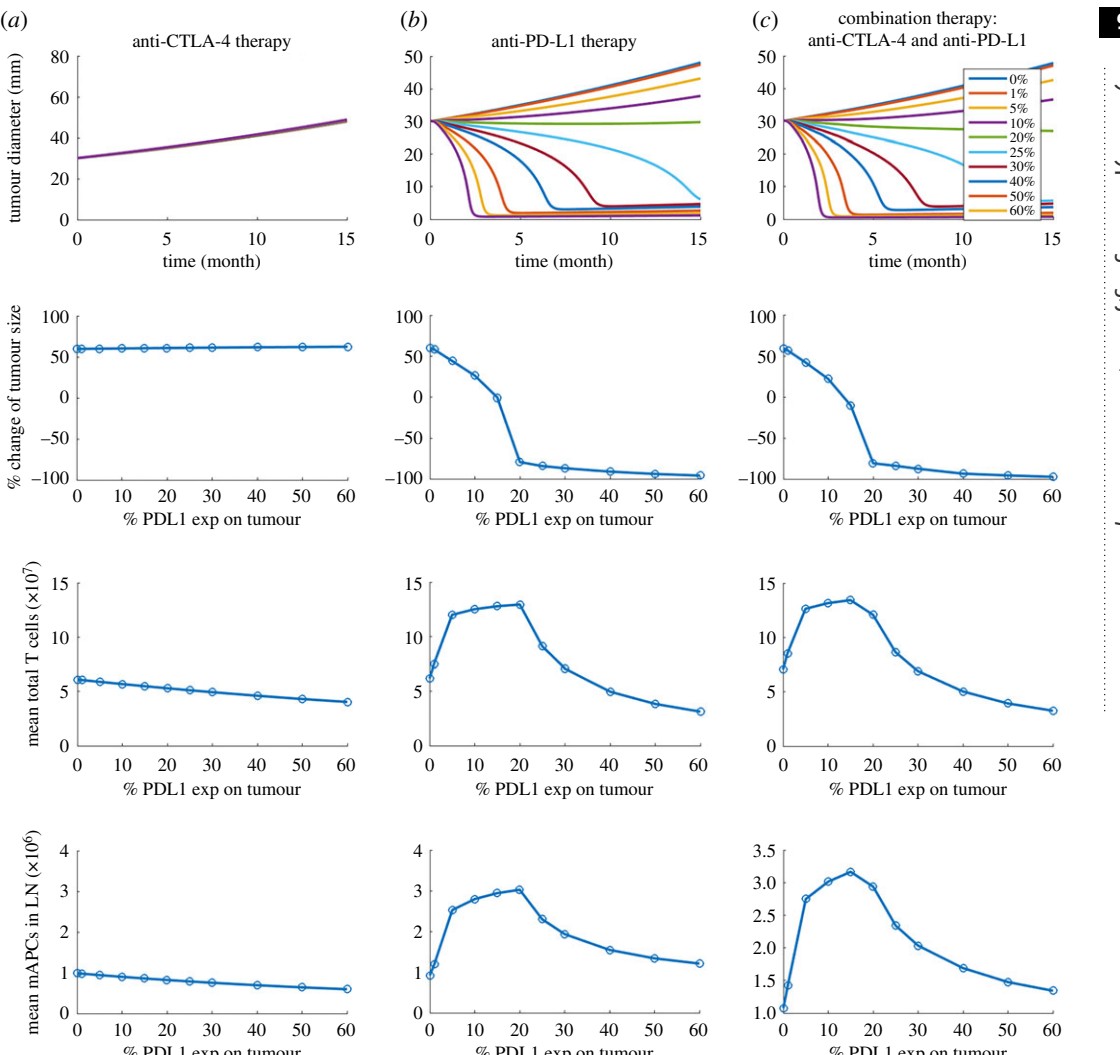

**Figure 4.** $(a-c)$ Time-dependent tumour diameter, the percentage change of end tumour size, mean total T cells and mean mAPC in the lymph node with different PD-L1 expressions on tumour cells.

administered monthly in anti-CTLA-4 therapy, four doses of 20 mg kg$^{-1}$ durvalumab are administered monthly followed by 10 mg kg$^{-1}$ every two weeks up to 15 months in anti-PD-L1 therapy. As shown in figure 3, as starting tumour size increases, the tumours progressively grow with anti-CTLA-4 monotherapy, while in response to anti-PD-L1 monotherapy and combination therapy tumours with initial size of 35 mm and smaller regress, whereas tumours with initial size above 40 mm grow. Notably, the response time of the smallest starting tumour size of 15 mm is slightly longer than that of 20 mm, possibly because a smaller tumour produces a smaller amount of neoantigen, which is vital for T-cell activation.

In figure 4, the PD-L1 expression on tumour cells is varied between 0 and 60%. Since experimental studies have shown that PD-L1 expression in early breast cancer is associated with higher T-cell infiltration, we assume that PD-L1-negative tumour cells are able to inhibit effector T cells by either the physical barrier of breast TME or non-PD-1/PD-L1 suppression pathways [40–42]. While no response is found in tremelimumab monotherapy, durvalumab therapy results in a complete response at the highest PD-L1 expression and tumour response becomes weaker as PD-L1 expression decreases. Next, we test the effect of antigen intensity in figure 5. While anti-CTLA-4 monotherapy at the selected dose and regimen still shows only minimal tumour response, anti-PD-L1 monotherapy and combination therapy suggest that the stronger the neoantigen is, the better the tumour response is.

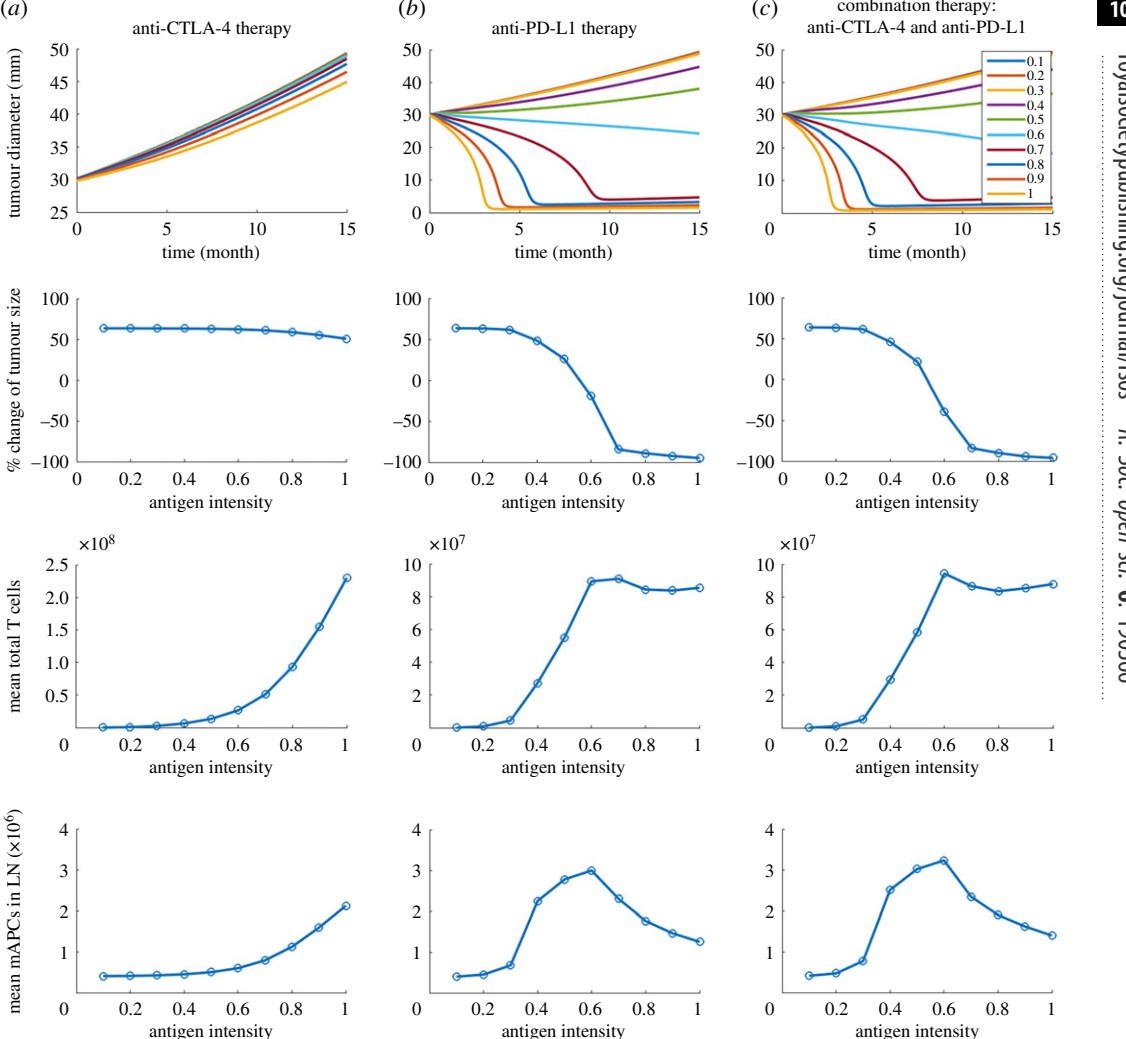

**Figure 5.** (a−c) Time-dependent tumour diameter, the percentage change of end tumour size, mean total T cells and mean mAPC in the lymph node with different antigen intensities.

## 3.3. Personalized prediction of tumour response to combination therapy

Now that we have investigated the correlations between the tumour response and each parameter of interest individually, we would like to simulate the combined effect of multiple parameters on each patient with ER+ and TNBC subtype in the clinical trial [20]. For each patient, we used the baseline tumour size and PD-L1 expression as initial conditions and for each subtype we use the average tumour growth rate and PD-L1 expression based on clinical data [43,44]. The contour plots in figure 6a,b show the relationship between the antigen intensity and PD-L1 expression and the tumour response to combination therapy, demonstrated by the percentage change of tumour size at the end of simulations, at 15 months. Both high PD-L1 expression and strong antigen intensity are required to induce a partial response in both subtypes. In figure 6c,d, the relationship between the antigen intensity and starting tumour size and the tumour response illustrates that antigen intensity is more critical to tumour response than the tumour size at the beginning of the therapy.

To further investigate the effect of antigen intensity, its threshold for partial response is plotted against PD-L1 expression in figure 7a. Not surprisingly, a higher antigen intensity is required for TNBC to have a partial response to combination therapy since TNBC has a higher tumour growth rate than ER+ breast cancer [44]. Similarly, antigen intensity threshold for partial response is plotted against starting tumour size (figure 7b). Interestingly, given the difference of tumour growth rate and PD-L1 expression between the two subtypes, the antigen intensity required for ER+ breast cancer is higher than that for TNBC and becomes similar to the starting tumour size increases. This result suggests that its high antigen intensity requirement may account for the fact that ER+ breast cancer has a lower response rate to immunotherapy due to its low immunogenicity [26].

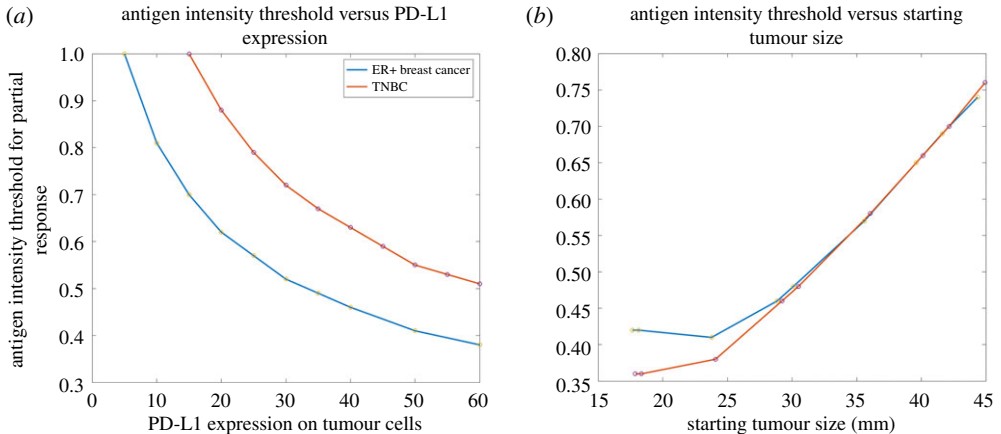

**Figure 6.** Percentage change of end tumour size versus antigen strength and (a,b) PD-L1 expression on tumour cells and (c,d) starting tumour size (with constant PD-L1 expressions of 33% and 59% for ER+ and TNBC, respectively). Simulations use the mean tumour growth rate for each subtype.

**Figure 7.** Thresholds of antigen strength for partial response (at least 30% decrease of tumour diameter) versus (a) PD-L1 expression and (b) starting tumour size for TNBC and ER+ breast cancer (with the mean PD-L1 expression on tumour cells and tumour growth rate for each subtype).

The clinical measurements of tumour size change [20] allow us to make our first attempt to make personalized predictions of tumour response to combination therapy. Four patients are selected from each subtype who have the most number of measurements available. First, we test whether a single factor can be used to recapitulate the heterogeneity of tumour response among the patients. The effect of each factor on tumour response is simulated by varying one factor at a time (electronic

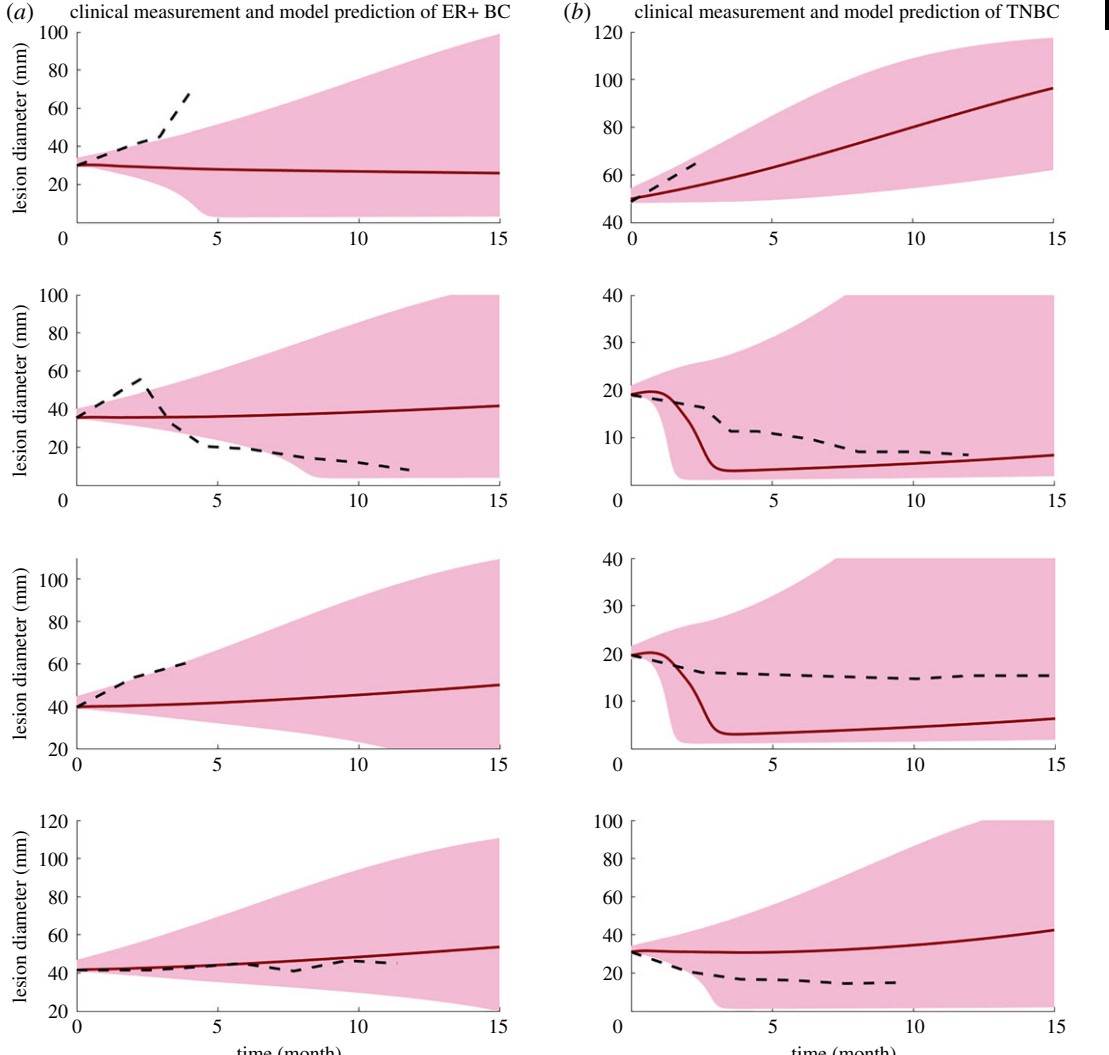

**Figure 8.** Clinical measurements and model prediction of ER+ breast cancer and TNBC. Dashed line represents the clinical measurement of the average lesion size of each patient. Solid lines represent the median model prediction with a range of prediction results (shaded area) using antigen strength of 0.4–0.6, PD-L1 expression of 20–40% on tumour cells for ER+ breast cancer and 40–60% for TNBC, assuming a uniform distribution for each parameter. Tumour growth rates are estimated by the mean tumour doubling time and kept within one standard deviation reported from clinical measurements for of 4 ER+ breast cancer patients (*a*) and 4 TNBC (*b*).

supplementary material, figures S2 and S4); the results demonstrate that any single factor above is not sufficient to describe the diversity of tumour response. In figure 8, black dash lines represent the average lesion size from clinical measurements, and the shade includes a range of prediction results. Since the antigen intensity and PD-L1 expression data are not available for each patient, we assume antigen intensity in the range 0.4–0.6 for both subtypes. The PD-L1 expression is assumed to be 20–40% and 40–60% for ER+ and TNBC, respectively, estimated based on human data [43]; the tumour growth rate of each subtype is estimated using the mean tumour doubling time and standard deviation from clinical measurements [44]. Here we assume a uniform distribution for both PD-L1 expression and antigen intensity and keep tumour growth rate within one standard deviation. Alternatively, we further assume a normal distribution for PD-L1 expression and antigen intensity with the same mean values and estimate their standard deviations so that the areas under the curves are the same as uniform distributions within their ranges. Figure 9 demonstrates the prediction results with 95%, 65% and 35% confidence intervals. Both figures represent an admittedly rough and preliminary prediction of individual patients' response to combination therapy, with the red line showing the median model prediction. While the clinical measurements of tumour size mostly fall into the range of our prediction results and the trends are instructive, the median predicted tumour size changes of TNBC patients are stronger than the clinical results, especially at large starting tumour sizes.

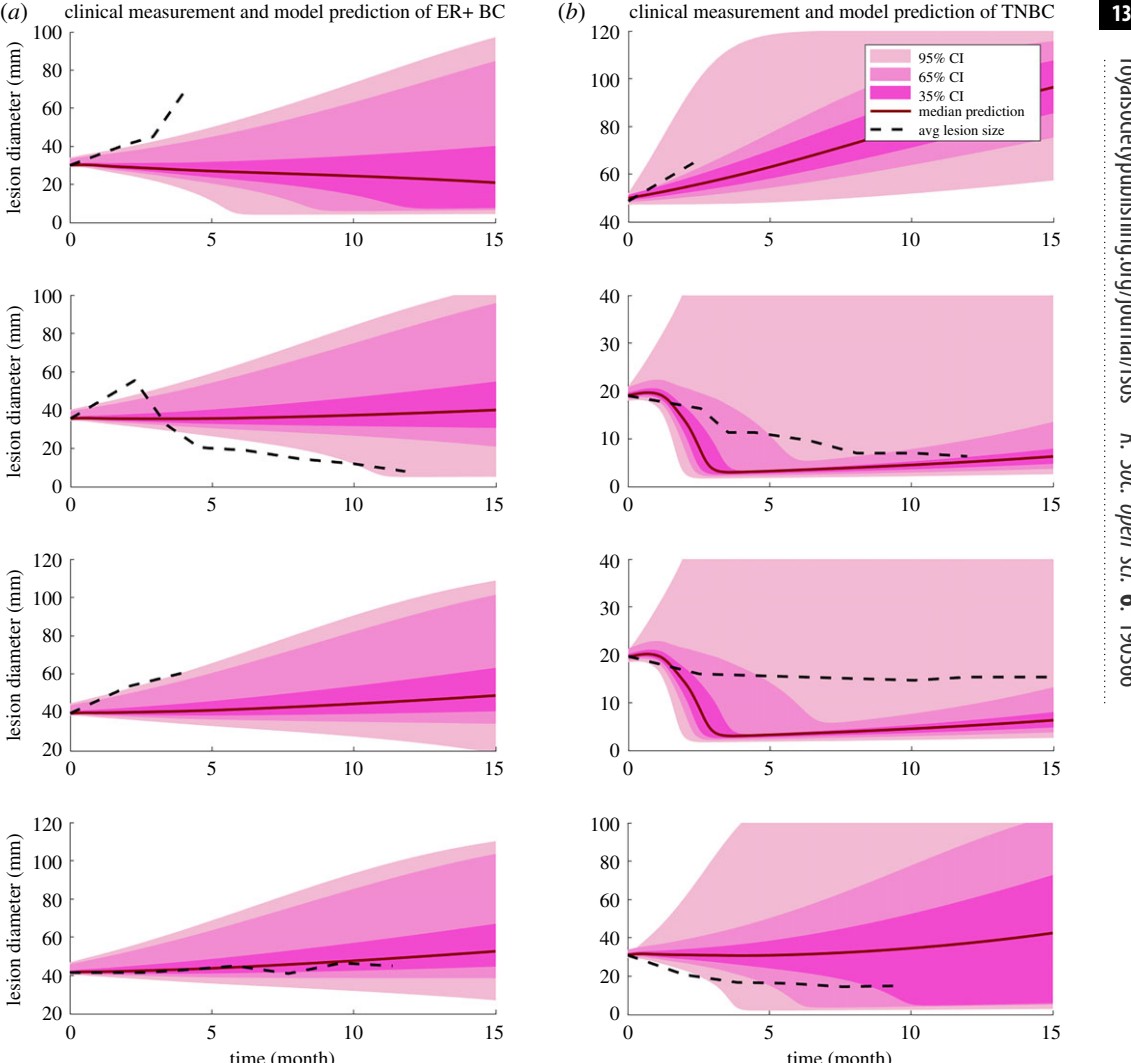

**Figure 9.** Clinical measurements and model prediction of ER+ breast cancer and TNBC. Dashed line represents the clinical measurement of the average lesion size of each of 4 ER+ breast cancer patients (*a*) and 4 TNBC (*b*). Solid lines represent the median model prediction with 95%, 65% and 35% confidence intervals assuming a normal distribution for antigen strength of 0.4–0.6, PD-L1 expression of 20–40% on tumour cells for ER+ breast cancer and 40–60% for TNBC. The standard deviations are estimated by assuming the same area under curve as the uniform distribution within the range. Parameters are kept within 1.96, 0.936 and 0.454 standard deviations corresponding to confidence intervals of 95%, 65% and 35%, respectively.

We further perform a Latin hypercube sampling and partial rank correlation coefficient (LHS\PRCC) analysis using a sample size of 1000 to examine the uncertainty and sensitivity of predicted tumour response against potential biomarkers that have not been confirmed by clinical studies [45]. As shown in figure 10, tumour response to anti-CTLA-4 and anti-PD-L1 therapy is sensitive to effector T-cell apoptosis, Treg/MDSC level in the tumour, T-cell clonality and the effect of checkpoint expression by Tregs on effector T cells and mAPCs, in addition to the parameters discussed above.

## 4. Discussion

The heterogeneity of response to checkpoint blockade therapy among the subtypes of breast cancer has highlighted the need for predictive biomarkers to identify responders. As single-agent monotherapies result in limited efficacy among most of the subtypes, combination strategies will be required [2]. While a number of ongoing clinical trials are investigating the efficacy of combination therapies in breast cancer and the correlation between responders and potential biomarkers, the large number of

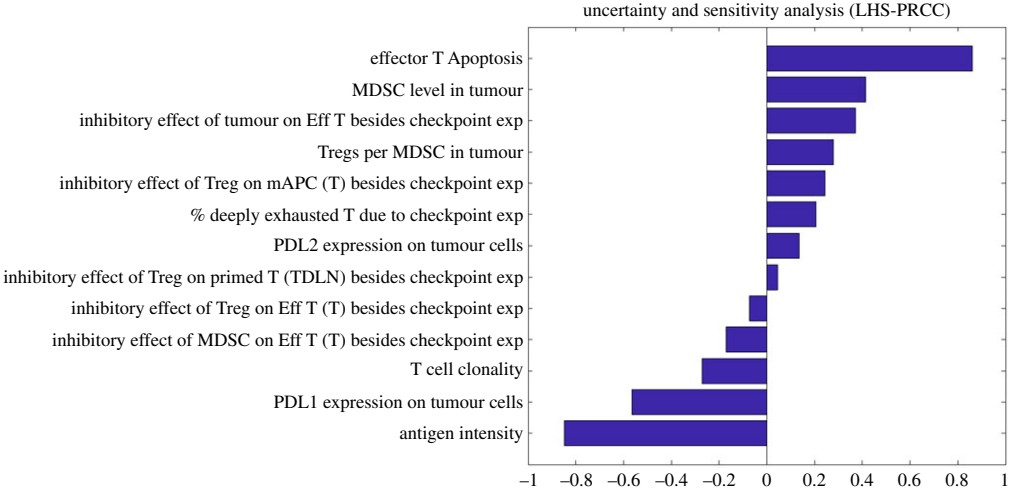

**Figure 10.** Uncertainty and sensitivity analysis of percentage tumour size change.

possible targets, biomarkers and cancer subtypes, and high cost of these trials all underscore the need for computational models that can describe the complexity of physiological processes in patients and mimic their response to mono- and combination therapies. Recently, several models have been developed that focus on the tumour microenvironment to investigate the immune evasion mechanisms and how blockade agents interfere with them [46–53]. However, the effects of immune checkpoint blockade in tumour-draining lymph nodes have not been elucidated, and the amount of molecular and cellular mechanistic detail is not as comprehensive as in the present model. As tumour cells are not intrinsically resistant to immune response, the proposed model aims to analyse the immune suppression and evasion in TDLNs and TME, both of which are important for tumour progression [54]. The present model integrates the processes involved in the patient, including APC maturation, T-cell activation, T-cell trafficking and tumour rejection, which allows us to investigate the effects of several factors such as pretreatment tumour size, tumour growth rate, PD-L1 expression and antigen intensity on a more comprehensive scale.

Starting from factors including the composite tumour burden, PD-L1 expression and antigen intensity, we investigate correlations between each factor and the tumour response to monotherapy and combination therapy. By changing one factor at a time, we confirm that each factor is critical to inducing tumour response to both anti-PD-L1 monotherapy and combination therapy using characteristics of an average breast cancer patient. However, it is difficult to conclude whether the starting tumour size has a strong correlation with tumour response to immunotherapy as small tumour size is unlikely to trigger enough T-cell production for tumour eradication; this question requires further investigation, especially since there is evidence that pretreatment tumour size correlates poorly with treatment outcome in the metastatic setting [55,56]. Further, clinically the high response rate is not always observed among patients with high neoantigen load, PD-L1 expression or antigen intensity, and the correlations between each factor and the tumour response are also different among the different subtypes of breast cancer [2]. For example, the majority of patients in the PD-L1 positive cohort do not respond to the blockade therapy even though they have higher ORR than the PD-L1-negative cohort, possibly due to the immune suppression in TDLN or other resistance mechanisms [57]. Thus, we further investigate the combined effects of more than one factor in tumour response in each subtype.

According to model-based analysis, the tumour response to combination therapy is highly sensitive to both PD-L1 expression on tumour cells and antigen intensity. A higher PD-L1 expression and antigen intensity are required by TNBC patients than ER+ breast cancer patients due to the higher tumour growth rate of TNBC; the relatively high response rate of TNBC to immunotherapy may be related to its relatively higher PD-L1 expression on average [43]. Notably, the model assumes that PD-L1-negative tumour cells are able to inhibit the infiltration and function of a majority of effector T cells through physical barrier or other pathways, which accounts for the inhibition of TIL recruitment. In addition, the calculated antigen intensity threshold for partial response suggests that a higher antigen intensity is required for ER+ breast cancer patients with tumour size smaller than 30 mm, which may account for the fact that lower ORRs are observed in clinical trials on ER+ breast cancer [58]. In fact,

the low immunogenicity of ER+ breast cancer has been recognized to be a reason for the low immune activation in TDLN, which leads to an immune-suppressive TME [59].

In addition, we present personalized simulations of tumour response to combination therapy using a range of parameters, within the limitations of the available data from the pilot clinical trial [20]. By controlling the PD-L1 expression, antigen intensity and tumour growth rate within a reasonable range based on published clinical data, we find that these three factors are able to describe the tumour response to combination therapy of both subtypes. Notably, an overestimation of tumour response is observed in TNBC patients with tumour size smaller than 20 mm. While a smaller tumour size with high PD-L1 expression is likely to have a complete response based on model predictions, the clinical measurements show only partial responses. This phenomenon suggests that there may exist additional resistance mechanisms in breast TME or TDLNs that are not incorporated into the model. Furthermore, the small sample size included in the clinical study and our selection of patients with the most clinical measurements for personalized prediction may lead to a bias towards the responders. However, due to the lack of clinical measurement of PD-L1 expression and other clinical data, most of the parameters are set to be the average value among all cancer patients, which limits a more comprehensive investigation of potential biomarkers. Due to the small number of patients and measurements of tumour size in this pilot trial, the distribution of parameters of interest cannot be determined and they are assumed to have normal distributions within their physiologically reasonable ranges for personalized simulations. By focusing on the antibody drugs, dose regimen and the available measurements in this particular pilot trial, we believe that with an extensive sensitivity analysis the model allows us to investigate the characteristics and resistance mechanisms in breast cancer that affect the therapy outcome.

In the current model, we focus on the immunosuppression in TDLNs and TME due to the inhibition of host immune response by Tregs and MDSCs, and their inhibitory effect on both T cells and APCs are heavily dependent on checkpoint expression. As a result, simulated checkpoint blockade therapy can result in a robust immune response that leads to a complete response in patients with high PD-L1 expression and antigen intensity. Although this result aligns with the strong correlation between the tumour response and the factors above reported by clinical studies, the additional resistance mechanisms that are independent of checkpoint expression should be incorporated into the model. An important factor that should be considered is the effects of heterogeneous expression of cytokines among patients. As suggested by the global sensitivity analysis (figure 10), effector T-cell apoptosis, induced by IL-10 secretion from MDSCs and tumour cells, inhibition of cytotoxic activity of effector T cells through arginase or NO production by MDSCs that do not require antigen-specific interactions, and Treg induction and expansion by IL-10 and TGF-β may also contribute to the immunosuppressive TME in breast cancer patients [60–62]. MDSC level in each breast cancer subtype and the inhibitory effect of checkpoint expression by Tregs/MDSCs on effector T cells and mAPCs are also needed to be quantitatively determined by experiments. In fact, the MDSC level has shown a significant negative correlation with the production of pro-inflammatory cytokines and the overall survival of patients with breast cancer [63].

Although the model aims to focus on a metastatic setting, the difference in metastatic site, agents used in previous chemotherapy, and their unknown effects on immune system among patients make it difficult to estimate their effects on the efficacy of adjuvant checkpoint blockade therapy. The pharmacokinetics of the antibody may also vary due to the difference in permeability and surface area between blood capillaries and metastatic lesions. In addition, the model has only one type of neoantigen-specific effector T cell produced from TDLNs that has a constant binding affinity with neoantigens, defined by a normalized antigen intensity, due to the lack of clinical measurements of tumour-specific effector T-cell level and the binding affinity of TCR with neoantigen peptides in breast cancer. As more data become available, a detailed antigen presentation mechanism can be incorporated into the model. Importantly, the model provides a platform that can be adapted to other therapies to investigate synergies with the checkpoint blockade therapy [64]. In recent studies, entinostat treatment, which significantly reduces circulating MDSC levels, demonstrates a strong correlation with overall survival rate of breast cancer patients and is involved in ongoing clinical trials with immune checkpoint blockade. The addition of cytokine expression and MDSCs in the future version of the model would allow us to investigate tumour response to the recent therapies in advanced breast cancer [65].

In addition to our proposed ODE-based model, there are other ODE-based and agent-based models that aim to predict the efficacy of cancer vaccines and small-molecule modulators of cancer immunity [66–69]. Based on recent success of preclinical studies of epigenetic modulators in breast cancer these

models can be adapted to predict optimal dosing schedules. Although our proposed model demonstrated its potential to predict the tumour response given the distribution of physiological parameters at population level, one of the challenges of ODE-based models is to capture the heterogeneity of tumour microenvironment. Although we capture the differential checkpoint expression on tumour cells by dividing them into different subtypes, the number of ligands and receptors on cell surfaces are assumed to be constant for the checkpoint-positive subtypes. To account for all the possible differential expression of checkpoints within the same cell type requires a much greater number of parameters and equations, which increases the model complexity and data requirement for model calibration. For the same reason, the present model only includes major species of interest in TME, including T-cell subtypes (Teff and Tregs), APCs, MDSCs and tumour cells; the modelling of spatial distribution of cells, immune repertoire and their specificity requires a different approach, e.g. agent-based modelling [70].

## 5. Conclusion

The proposed quantitative systems pharmacology model integrates relevant immune and cancer-related compartments and processes of individual breast cancer patients and specifies a number of immune suppression and evasion mechanisms in both TDLN and TME. It aims to identify potential biomarkers, additional resistance mechanisms, and provide a basis for future predictions of tumour response to immunotherapy for cohorts of patients and eventually for personalized therapy.

Data accessibility. The authors confirm that the data supporting the findings of this study are available within the article and its electronic supplementary material.

Authors' contributions. A.S.P. designed and supervised the project, revised the manuscript critically. O.M. built the model structure. H.W. modified the model, performed all simulations, analysed the simulation data and prepared a draft of the manuscript. I.H.B., P.V., B.W., R.N. and L.R. revised the manuscript critically. C.A.S.-M. provided clinical data, revised the manuscript critically. All authors have read and approved the final manuscript.

Competing interests. C.A.S.-M. receives research support from Pfizer and MedImmune and serves on the advisory board for Polyphor. I.H.B., P.V., B.W., R.N. and L.R. were employees of MedImmune. A.S.P. receives research support from MedImmune. Other authors declare that the research was conducted in the absence of any commercial or financial relationships that could be construed as a potential conflict of interest.

Funding. Supported by NIH grant R01CA138264 (A.S.P.) and a grant from MedImmune (A.S.P.).

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
