## [Reviewer comments · Royal Society Open Science]

Review History

RSOS-190366.R0 (Original submission)

Review form: Reviewer 1

Is the manuscript scientifically sound in its present form?

Yes

Are the interpretations and conclusions justified by the results?

Yes

Is the language acceptable?

Yes

Is it clear how to access all supporting data?

Yes

Do you have any ethical concerns with this paper?

No

Have you any concerns about statistical analyses in this paper?

No

Recommendation?

Major revision is needed (please make suggestions in comments)

Comments to the Author(s)

In this work, Wang and colleagues present an ordinary based equation system to investigate the relationship between the tumor response to checkpoint blockade therapy and antigen intensity, including their individual and combined effects on the immune system.

The work is well written and the results are clearly presented. All the equations are presented and one can be able to reproduce the work.

On the other hand, there are some critical points that need to be addressed before the manuscript is acceptable for publications.

Major points:

1) Authors failed to discuss other methodologies that were adopted to model the complex immune system behaviour both alone and in response to induced artificial immunity. There are a lot of these examples, especially in tumor immunology. Some are based on agent based modeling approaches. Others are based on complex differential equation systems, also based on enzymatic kinetics.

See for example:

*) Induction of T cell memory by a dendritic cell vaccine: a computational model. *Bioinformatics*, 30(13):1884–1891, 2014. (doi:10.1093/bioinformatics/btu059)

*) A Dynamic Model of Immune Responses to Antigen Presentation Predicts Different Regions of Tumor or Pathogen Elimination, *Cell Systems*, Volume 4, Issue 2, 2017, 231-241. doi:10.1016/j.cels.2016.12.003

*) In silico modeling and in vivo efficacy of cancer-preventive vaccinations. *Cancer Research*, 70(20):7755–7763, 2010. (doi:10.1158/0008-5472.CAN-10-0701)

*) Computational modeling reveals MAP3K8 as mediator of resistance to vemurafenib in thyroid cancer stem cells. *Bioinformatics*, 2018. (doi:10.1093/bioinformatics/bty969)

It is useful if the authors add to the discussion session a paragraph or two reviewing these important works in the field.

2) Authors failed to discuss the advantage and the disadvantages of their methodology. ODE systems are good when dealing with populations. But they fail to address important aspects when immune system is modelled. For example, specificity and immune system repertoire are not taken into account. Moreover, different types of lymphocytes subpopulation are not considered at all. See for example CD4(Th1, Th2, Th17), CD8. Authors should mention these in the discussion session.

Decision letter (RSOS-190366.R0)

08-Apr-2019

Dear Mr Wang,

The editors assigned to your paper ("In silico simulation of a clinical trial with anti-CTLA-4 and anti-PD-L1 immunotherapies in breast cancer using a systems pharmacology model") have now received comments from reviewers. We would like you to revise your paper in accordance with the referee and Associate Editor suggestions which can be found below (not including confidential reports to the Editor). Please note this decision does not guarantee eventual acceptance.

Please submit a copy of your revised paper before 01-May-2019. Please note that the revision deadline will expire at 00.00am on this date. If we do not hear from you within this time then it will be assumed that the paper has been withdrawn. In exceptional circumstances, extensions may be possible if agreed with the Editorial Office in advance. We do not allow multiple rounds of revision so we urge you to make every effort to fully address all of the comments at this stage. If deemed necessary by the Editors, your manuscript will be sent back to one or more of the original reviewers for assessment. If the original reviewers are not available, we may invite new reviewers.

- Data accessibility

If you wish to submit your supporting data or code to Dryad (<http://datadryad.org/>), or modify your current submission to dryad, please use the following link:
<http://datadryad.org/submit?journalID=RSOS&manu=RSOS-190366>

- **Competing interests**

- **Authors' contributions**

- **Acknowledgements**

- **Funding statement**

on behalf of Dr Marco Viceconti (Associate Editor) and Professor Pietro Cicuta (Subject Editor)
openscience@royalsociety.org

Comments to Author:

Reviewers' Comments to Author:

Reviewer: 1

Comments to the Author(s)

In this work, Wang and colleagues present an ordinary based equation system to investigate the relationship between the tumor response to checkpoint blockade therapy and antigen intensity, including their individual and combined effects on the immune system.

The work is well written and the results are clearly presented. All the equations are presented and one can be able to reproduce the work.

On the other hand, there are some critical points that need to be addressed before the manuscript is acceptable for publications.

Major points:

1) Authors failed to discuss other methodologies that were adopted to model the complex immune system behaviour both alone and in response to induced artificial immunity. There are a lot of these examples, especially in tumor immunology.

Some are based on agent based modeling approaches. Others are based on complex differential equation systems, also based on enzymatic kinetics.

See for example:

*) Induction of T cell memory by a dendritic cell vaccine: a computational model. *Bioinformatics*, 30(13):1884-1891, 2014. (doi:10.1093/bioinformatics/btu059)

*) A Dynamic Model of Immune Responses to Antigen Presentation Predicts Different Regions of Tumor or Pathogen Elimination, *Cell Systems*, Volume 4, Issue 2, 2017, 231-241. doi:10.1016/j.cels.2016.12.003

*) In silico modeling and in vivo efficacy of cancer-preventive vaccinations. *Cancer Research*, 70(20):7755-7763, 2010. (doi:10.1158/0008-5472.CAN-10-0701)

*) Computational modeling reveals MAP3K8 as mediator of resistance to vemurafenib in thyroid cancer stem cells. *Bioinformatics*, 2018. (doi:10.1093/bioinformatics/bty969)

It is useful if the authors add to the discussion session a paragraph or two reviewing these important works in the field.

2) Authors failed to discuss the advantage and the disadvantages of their methodology. ODE systems are good when dealing with populations. But they fail to address important aspects when immune system is modelled. For example, specificity and immune system repertoire are not taken into account. Moreover, different types of lymphocytes subpopulation are not considered at all. See for example CD4(Th1, Th2, Th17), CD8.

Authors should mention these in the discussion session.

Author's Response to Decision Letter for (RSOS-190366.R0)

See Appendix A.

Decision letter (RSOS-190366.R1)

24-Apr-2019

Dear Mr Wang,

I am pleased to inform you that your manuscript entitled "In silico simulation of a clinical trial with anti-CTLA-4 and anti-PD-L1 immunotherapies in breast cancer using a systems pharmacology model" is now accepted for publication in Royal Society Open Science.

on behalf of Dr Marco Viceconti (Associate Editor) and Pietro Cicuta (Subject Editor)
openscience@royalsociety.org

Appendix A

09-Apr-2019

Dear Editors:

Thank you for your email dated 8 Apr 2019 enclosing the reviewer's comments. We have carefully reviewed the comments and have revised the manuscript accordingly. Our responses are given below. Changes to the manuscript are made using track changes mode.

We hope the revised version is now suitable for publication and look forward to hearing from you.

Sincerely,

Hanwen Wang

hwang163@jhu.edu

Response to Reviewer 1:

We thank the reviewer for the review of our paper and constructive comments. We have answered each point below.

1. Authors failed to discuss other methodologies that were adopted to model the complex immune system behaviour both alone and in response to induced artificial immunity. There are a lot of these examples, especially in tumor immunology.

Some are based on agent based modeling approaches. Others are based on complex differential equation systems, also based on enzymatic kinetics.

See for example:

*) Induction of T cell memory by a dendritic cell vaccine: a computational model. *Bioinformatics*, 30(13):1884–1891, 2014. (doi:10.1093/bioinformatics/btu059)

*) A Dynamic Model of Immune Responses to Antigen Presentation Predicts Different Regions of Tumor or Pathogen Elimination, *Cell Systems*, Volume 4, Issue 2, 2017, 231-241. doi:10.1016/j.cels.2016.12.003

*) In silico modeling and in vivo efficacy of cancer-preventive vaccinations. *Cancer Research*, 70(20):7755–7763, 2010. (doi:10.1158/0008-5472.CAN-10-0701)

*) Computational modeling reveals MAP3K8 as mediator of resistance to vemurafenib in thyroid cancer stem cells. *Bioinformatics*, 2018. (doi:10.1093/bioinformatics/bty969)

It is useful if the authors add to the discussion session a paragraph or two reviewing these important works in the field.

Response: We added a paragraph regarding the other model types in tumor immunology, including the suggested examples, and their potential applications in the Discussion section (please see the last paragraph in Discussion, page 13).

2. Authors failed to discuss the advantage and the disadvantages of their methodology. ODE systems are good when dealing with populations. But they fail to address important aspects when immune system is modelled. For example, specificity and immune system repertoire are not taken into account. Moreover, different types of lymphocytes subpopulation are not considered at all. See for example CD4(Th1, Th2, Th17), CD8. Authors should mention these in the discussion session.

Response: The limitations of ODE-based models are also added to the last paragraph in the Discussion section. Although we tried to capture the heterogeneity of tumor microenvironment by adding tumor species expressing different types of checkpoint ligands and receptors, the modeling of spatial distribution and the immune repertoire may require other approaches such as agent-based modeling. Also for simplicity, only the major species of interest are included in the current model (please see the last paragraph in the Discussion, page 13).